# Transcriptome Analysis Reveals Brassinolide Signaling Pathway Control of Foxtail Millet Seedling Starch and Sucrose Metabolism under Freezing Stress, with Implications for Growth and Development

**DOI:** 10.3390/ijms241411590

**Published:** 2023-07-18

**Authors:** Xiatong Zhao, Ke Ma, Zhong Li, Weidong Li, Xin Zhang, Shaoguang Liu, Ru Meng, Boyu Lu, Xiaorui Li, Jianhong Ren, Liguang Zhang, Xiangyang Yuan

**Affiliations:** College of Agronomy, Shanxi Agricultural University, Taigu, Jinzhong 030801, China; b20191017@stu.sxau.edu.cn (X.Z.); b20223010009@cau.edu.cn (K.M.); li-s20212189@stu.sxau.edu.cn (Z.L.); z20213117@stu.sxau.edu.cn (W.L.); zx1150472698@163.com (X.Z.); 15386943302@163.com (S.L.); s20212192@stu.sxau.edu.cn (R.M.); 18735424089@163.com (B.L.); lixiaorui@sxau.edu.cn (X.L.); nwafu_rjh@163.com (J.R.); zhangliguang1982@126.com (L.Z.)

**Keywords:** foxtail millet, freezing stress, RNA-seq, starch metabolism, sucrose metabolism, brassinolide signaling pathway

## Abstract

Low-temperature stress limits the growth and development of foxtail millet. Freezing stress caused by sudden temperature drops, such as late-spring coldness, often occurs in the seedling stage of foxtail millet. However, the ability and coping strategies of foxtail millet to cope with such stress are not clear. In the present study, we analyzed the self-regulatory mechanisms of freezing stress in foxtail millet. We conducted a physiological study on foxtail millet leaves at −4 °C for seven different durations (0, 2, 4, 6, 8, 10, and 12 h). Longer freezing time increased cell-membrane damage, relative conductance, and malondialdehyde content. This led to osmotic stress in the leaves, which triggered an increase in free proline, soluble sugar, and soluble protein contents. The increases in these substances helped to reduce the damage caused by stress. The activities of superoxide dismutase, peroxidase, and catalase increased reactive oxygen species (ROS) content. The optimal time point for the response to freezing stress was 8 h after exposure. The transcriptome analysis of samples held for 8 h at −4 °C revealed 6862 differentially expressed genes (DEGs), among which the majority are implicated in various pathways, including the starch and sucrose metabolic pathways, antioxidant enzyme pathways, brassinolide (BR) signaling pathway, and transcription factors, according to Gene Ontology (GO) and Kyoto Encyclopedia of Genes and Genomes (KEGG) enrichment. We investigated possible crosstalk between BR signals and other pathways and found that BR signaling molecules were induced in response to freezing stress. The beta-amylase (BAM) starch hydrolase signal was enhanced by the BR signal, resulting in the accelerated degradation of starch and the formation of sugars, which served as emerging ROS scavengers and osmoregulators to resist freezing stress. In conclusion, crosstalk between BR signal transduction, and both starch and sucrose metabolism under freezing stress provides a new perspective for improving freezing resistance in foxtail millet.

## 1. Introduction

Low-temperature stress in plants can be categorized into two types: chilling stress (>0 °C) and freezing stress (<0 °C). These stressors can have detrimental effects on plant survival, growth, and production capacity [1,2,3]. Owing to the static nature of plants, they are not equipped to escape adversity; however, many have evolved developmental plasticity to adjust to changing environmental conditions [4]. During crop growth, exposure to cold temperatures (10 °C) in early spring can lead to yield loss [5]; however, abrupt freezing stress in early autumn and late spring can result in even more significant economic losses [6]. This highlights the sensitivity of crops to low temperatures during the vegetative growth stage [7,8]. Effective control of diseases and pests in crops often involves chemical regulation or cultivation measures. In contrast, sudden-onset low temperatures cannot be prevented. Therefore, genetic improvement in freezing resistance is considered the most promising approach to address this issue [9].

To adapt to freezing stress, plants utilize various endogenous strategies to enhance freezing resistance, including (1) the accumulation of osmotic substances to prevent water loss and protect macromolecular structures in cells [10]; (2) the production of ROS, including sugars, which are fine-tuned by the plant’s antioxidant scavenging system to relieve freezing damage [11] (in addition to enzymatic and non-enzymatic processes, sugar plays a crucial role as an emerging ROS scavenger [12]); and (3) the promotion of prohormone biosynthesis, which amplifies the stress signal and induces stress-related genes to respond via hormone signaling molecules such as abscisic acid (ABA), auxin (IAA), and BR, helping to resist abiotic stress [13,14].

Genome-wide transcriptome profiling is an effective method for identifying genes of interest by detecting and elucidating molecular mechanisms involved in physiological processes [15]. In recent years, low-temperature-responsive transcriptomes have been extensively studied in various crops, leading to an improved understanding of the underlying molecular networks. For instance, a recent study revealed the upregulation of BR biosynthetic genes, such as *EnDWARF* and *EnCYP72A14*, in response to low-temperature stress [16]. Another study demonstrated that exogenous BR treatment with barrel medicine (*Medicago truncatula* L.) enhanced the expression levels of CBF and COR genes, including *mtaco1*, *mtacs2*, and *mtacs7*, ultimately activating the defense system and enhancing low-temperature tolerance [17]. Studies have demonstrated that brassinosteroids can promote the rapid expression of certain genes related to low-temperature tolerance [18,19]. Carbohydrates play crucial roles in providing carbon and energy for cell building and metabolic regulation. This regulation is particularly important in the earliest phases of development for controlling metabolism, resisting stress, and regulating growth and development [20]. Studies have also shown that sucrose can counteract the detrimental effects of low-temperature stress on enzyme activity by increasing levels of metabolites via various pathways [21,22].

Foxtail millet (*Setaria italica* (L.) Beauv.) is one of the world’s oldest crops, domesticated in China 8700 years ago. It is a new model species among C_4_ grass crops and provides 6 million tons of food throughout Southern Europe and Asia, ranking second in the world’s total cereal production [23,24,25]. This species plays a vital role in adjusting crop planting structures in China. In recent years, many crops have reported multiple responses to low-temperature stress, with spring-sown crops mainly focusing on cold stress [26,27] and freezing stress caused by sudden drops in temperature during critical periods of growth and development, such as late spring [28,29]. To cope with gradually decreasing low-temperature environments, winter-sown crops improve their cold resistance with cold adaptation. However, spring-sown crops do not experience autumn dormancy, cold adaptation, or low-temperature exercise, making them susceptible to abrupt freezing stress that can cause fatal damage [30]. Foxtail millet production requires the selection of appropriate varieties based on local climatic conditions to ensure both quality and yield. However, it is important to consider the ability of a variety to withstand abrupt freezing stress. Currently, the ability of millet to cope with sudden temperature drops and the appropriate response strategies are not well understood. This study aimed to analyze changes in foxtail millet in the seedling stage, both physiologically and in terms of gene expression, following a sudden cooling treatment. The results of this study provide a valuable reference for selecting foxtail millet planting varieties in areas prone to freezing stress, implementing effective management strategies to minimize the damage caused by freezing stress, and developing future research on how to deal with such unfavorable environmental conditions.

## 2. Results

### 2.1. Seedling Survival Rate of Foxtail Millet Leave during Freezing Stress

The growth changes of foxtail millet seedlings under −4 °C freezing stress for up to 12 h were observed. There were significant differences in the leaf morphological changes of H among the different treatments (Figure 1). Under freezing stress for 6 h, the survival rate of H decreased significantly, by 37.3%, compared with 0 h. Under freezing stress at −4 °C for 10 and 12 h, severe freezing damage occurred to leaves and stems, which showed dehydration and wilting; the survival rates were just 16.4% and 4.8%, respectively. After 8 h, the growth state of H was poor, and the survival rate was low. After 12 h, the plant height of H was consistent with normal growth under freezing stress, with no significant changes.

### 2.2. Antioxidant Enzymatic Activities of Foxtail Millet under Freezing Stress

To study the response of foxtail millet cells to growth, oxidative, and osmotic damage under freezing stress, we measured superoxide dismutase (SOD), peroxidase (POD), and catalase (CAT) activities; free proline content; soluble protein content; soluble sugar content; relative electrical conductivity (REC), malondialdehyde (MDA) content; and plant height of foxtail millet plant leaves (Figure 2, Figure 3 and Figure 4). The activities of SOD, POD, and CAT enzymes in H showed an increasing trend with the extension of stress time; all reached a peak at 12 h, at which point they were significantly increased by 181.9%, 116.0%, and 90.6, respectively, compared with 0 h (Figure 2).

### 2.3. Changes in Osmotic Adjustment of Foxtail Millet under Freezing Stress

As shown in Figure 3A, soluble sugar content in H began to increase significantly after 6 h of −4 °C freezing stress and high soluble sugar content was maintained from 6 to 12 h. Soluble sugar content increased from 3.25 mg·g FW^−1^ (0 h) to 7.53 mg·g FW^−1^ (12 h). Soluble protein content increased from 3.96 mg·g FW^−1^ (0 h) to 12.53 mg·g FW^−1^ (12 h) (Figure 3B). Under freezing stress for 8 h, the proline value of H was 501.54 µg·g FW^−1^, which was a significant increase, 91.2%, compared with 0 h, and there was an increase of 273.15 µg·g FW^−1^ at 12 h compared with 8 h (Figure 3C).

### 2.4. Membrane Injury and Chloroplast Ultrastructures of Foxtail Millet under Freezing Stress

As shown in Figure 4A, the REC of H leaves increased rapidly under freezing stress for 8 h. MDA content was consistent with the trend of relative conductivity, with a relatively small increase (Figure 4B). To explore the mechanism of freezing damage, we set up three biological replicates and selected leaf samples at 0 h as the control group (N) and at 8 h as the treatment group (C) for transcriptome sequencing. Electron microscopy observations of foxtail millet leaves (Figure 4C) revealed that freezing stress resulted in a significant accumulation of starch grains in the chloroplasts of H.

### 2.5. Transcriptome Sequencing and Correlation Analysis

This study analyzed the changes in the transcription levels of foxtail millet seedlings under freezing stress using the Illumina HiSeq 2500 platform. RNA samples were collected from both normal (N) and frozen (C) leaves that were exposed to −4 °C for 8 h. Two cDNA libraries were constructed for RNA-Seq analysis. A total of 62.14 GB of clean data were obtained from the cDNA libraries, with each individual sample reaching 8.50 GB. The quality of the data was high, with a Q30 base percentage of >94.06% and consistent GC content of approximately 54%. Clean reads were mapped to the foxtail millet reference genome using HISAT2. The percentage of clean reads mapped to the reference genome was between 94.5% and 95.2%. Of the clean reads, 92% could be mapped unambiguously and were used for further analysis (Table 1). The correlation of three biological replicates at each time point was used to score the samples to ensure reliability. The correlation coefficients were calculated from the fragments per kilobase per million mapped (FPKM) values obtained for each sample. The results showed that the normal biological replicates (N) had a correlation coefficient of ≥0.96, while the freeze-treated (C) ones had a correlation coefficient of ≥0.94 (Appendix A). These correlations indicate the reliability of the experimental samples for further analysis.

### 2.6. Defining DEGs

FPKM was used to normalize the 37,479 genes from the mapped libraries. In total, 6862 DEGs were identified between N and C (Figure 5A,B) [31]. All DEGs were subjected to hierarchical clustering to observe gene expression patterns assessed with the log_10_FPKM values of the two groups (Figure 5C). Group C had 3577 upregulated and 3285 downregulated genes compared with the N group. These results indicate that freezing stress significantly affects the transcription of a subset of genes in response to normal conditions.

### 2.7. GO and KEGG Analyses of Freezing Stress DEGs

GO enrichment analyses revealed the biological processes, molecular functions, and cellular component categories associated with the 6862 DEGs in the two groups (Figure 6A). Most transcripts in the BP category were enriched in metabolic processes (GO:0008152), cellular processes (GO:0009987), single-organism processes (GO:0044699), biological regulation (GO:0065007), and response to stimulus (GO:0050896). Enrichment was observed in genes related to cellular components (CCs), such as cells (GO:0005623), cell parts (GO:0044464), organelles (GO:0043226), membranes (GO:0016020), and membrane parts (GO:0044425). Binding (GO:0005488), catalytic activity (GO:0003824), transporter activity (GO:005215), nucleic acid-binding transcription factor activity (GO:0001071), and structural molecule activity (GO:0005198) of MF were highly enriched.

To identify the active biological pathways of the DEGs in foxtail millet under freezing stress, we performed enrichment analysis by classifying them into KEGG biological pathways. Of the 2279 DEGs, 135 pathways were annotated. Among these, the top five pathways with the highest representation were “Plant hormone signal transduction (ko04075)”, “Plant–pathogen interaction (ko04626)”, “Starch and sucrose metabolismand (ko00500)”, “Endocytosis (ko04144)”, and “Glycerophospholipid metabolism (ko00564)”, with 205, 273, 102, 84, and 64 DEGs, respectively. Plant–pathogen interactions were mentioned in most DEGs, suggesting that plants subjected to freezing stress are more susceptible to pathogen infections. “Plant hormone signal transduction” was the second largest DEG category, which suggests that plant hormones play an important role in foxtail millet under freezing stress. The third-highest number of DEGs was found in the “Starch and sucrose metabolismand” category; this suggests that freezing stress may affect sugar metabolism in foxtail millet. This may be due to the need for sugar to provide energy during freezing stress resistance in foxtail millet. In addition, the “Glycerophospholipid metabolism” category indicated that fatty acid metabolism reacted well to freezing stress. A sizable number of biological processes in foxtail millet changed in response to freezing stress.

### 2.8. Starch and Sucrose Metabolism and the Antioxidant Defense Systema Related DEGs under Freezing Stress

The metabolism of starch and sucrose is crucial to plant response to freezing stress [32]. In the KEGG enrichment analysis, many DEGs were enriched, and all were upregulated. Many DEGs were upregulated under freezing stress, including starch, fructose, sucrose, glucose, and trehalose (Figure 4A). Sucrose synthase (SUS) and sucrose invertase (INV) play important roles in sucrose metabolism. Sugar precursors involved in sucrose metabolism must be broken down into hexoses such as glucose and fructose, or hexoses (such as UDP-glucose) must be differentiated by SUS or INV [33]. One SUS (Seita.9G410800) was considered a significant DEG. Six genes encoding trehalose 6-phosphate synthetase (TPS3) were also identified (Seita.1G347300, Seita.2G197800, Seita.5G305600, Seita.5G318400, Seita.6G144600, and Seita.6G166700).

Plant starch metabolism is regulated by a series of enzymes, among which amylase plays an important role in starch degradation. In this study, eight alpha-amylase (EC 3.2.1.1; AMY) genes (Seita.1G294400, Seita.7G178500, Seita.6G181600, Seita.5G434500, Seita.5G434200, Seita.5G434300, Seita.5G295100, and Seita.1G331000) and two β-amylases (EC 3.2.1.2; BAM; Seita.9G317500 and Seita.9G544600) participated in amylase degradation, all of which were upregulated (Figure 7). The contents of sucrose and starch, and β-amylase activity also increased significantly after freezing stress (Appendix A).

### 2.9. Brassinosteroids Signaling Pathway Related DEGs

The KEGG enrichment analysis showed that freezing stress in foxtail millet leaves significantly enriched genes related to hormone signal transduction (Figure 6B). As BRs play a crucial role in plant response to abiotic stress, we conducted an integrative analysis of changes in BR biosynthesis and signaling pathway gene expression, along with BR content. Among them, 25 DEGs were enriched in the term “brassinosteroids signaling pathway”. Under freezing stress, BRI (Seita.1G085300, Seita.1G085800, Seita.2G377800, Seita.3G240900, Seita.4G251500, Seita.5G432500, Seita.6G205500, Seita.9G227100, Seita.9G429400, Seita.J023100 and Setaria_itaica_newGene_3251) senses BR and binds to its extracellular domain, facilitating the binding of BRI1 to the co-receptor BAK1(Seita.1G077100, Seita.2G228400, Seita.4G060400, Seita.4G250600, Seita.5G092500, Seita.5G198300, Seita.7G095600, Seita.7G240400, Seita.5G016100, Seita.8G152200, Seita.9G060300) and initiating the activation of the BR signal. Downstream, BZR1/2 (Seita.5G143700) family members accumulate in a non-phosphorylated state and activate TCH4 (Seita.4G246400) to regulate plant growth and development, and their response to environmental stimuli (Figure 8). At the same time, the content of BRs increased significantly after 8 h under freezing stress at −4 °C compared with 0 h (Appendix A).

### 2.10. Transcription Factors under Freezing Stress

Among the 6862 DEGs, 1998 were identified as transcription factors (TFs) and were enriched in 65 transcription factor families. AP2-ERF, WRKY, bZIP, bHLH, MYB, NAC, and C2H2 were the five most abundant TFs identified (Figure 9).

### 2.11. Quantitative Real-Time PCR (qRT–PCR) Analysis

To validate the DEG data from RNA-Seq, 15 DEGs were randomly selected for qRT–PCR assays under freezing stress (Appendix A Appendix A). All of these genes were involved in the freezing response based on the enrichment analyses. They represent different functional categories or pathways, such as sugar metabolism, plant hormones, antioxidant enzymes, and TFs. This indicates that the RNA-Seq results obtained in this study are reliable (Figure 10).

## 3. Discussion

### 3.1. Effects of Freezing Stress on Membrane Injury of Foxtail Millet Leaves

The biological membrane system consists of cell membrane, nuclear membrane, and organelle membrane. The initial site of low-temperature damage occurs at the cell- and organelle-membrane levels, which can cause serious damage to their structure, function, stability, and enzyme activity. This can result in significant metabolic imbalance [34,35]. Under stress, increased REC indicates that plant membrane lipids are subjected to severe freezing stress. MDA is considered the end product of lipid peroxidation. The concentration of MDA in plant cells is an indicator of damage caused by membrane lipid peroxidation and the ability of cells to withstand freezing stress [36,37]. Deng et al. [38] showed that when Magnolia is subjected to freezing stress, severe lipid damage can lead to seedling death. This explains the increased seedling mortality observed at extremely low temperatures. Our results show that REC and MDA levels significantly increased after freezing stress, which may lead to a significant increase in seedling mortality owing to membrane lipid damage. As depicted in Appendix A, the “glycerophospholipid metabolism (ko00062)” pathway exhibited a significant upregulation of DEGs (Q-value = 0.053703). On the other hand, the DEGs of the “fatty acid elongation” pathway were significantly downregulated (Q-value = 0.353313), which suggests that the two metabolic pathways were differentially regulated under freezing stress. The expression of genes related to lipid metabolism in plants is believed to be linked to their response to low temperatures. This study identified differentially expressed genes involved in fatty acid elongation (ko00564) and glycerophospholipid metabolism (ko00062) (Figure 6B), which are crucial components of cell membranes. These metabolic processes could potentially enhance cell-membrane fluidity, serving as a mechanism for cell self-repair [39,40]. The damage caused by freezing stress to the leaves was significantly higher than the self-regulation capacity of foxtail millet leaves. This ultimately resulted in an increase in seedling mortality after being subjected to −4 °C treatment for 8 h.

### 3.2. Metabolic Pathways of Starch and Sucrose under Freezing Stress

Starch metabolism and sucrose metabolism play important roles in plant responses to freezing stress [32]. In KEGG enrichment analysis, many DEGs involving starch, fructose, sucrose, glucose, and trehalose were upregulated under freezing stress (Figure 4A). SUS and INV play important roles in sucrose metabolism. Sugar precursors involved in sucrose metabolism must be broken down into hexoses such as glucose and fructose, or hexoses (such as UDP-glucose) must be differentiated using SUS or INV [33]. The expression levels of one sucrose synthetase gene (seita.9G410800) and three sucrose invertase genes (seita.1G181700, seita.7G181300, and seita.9G102500) were significantly upregulated during freezing treatment. Plants accumulate trehalose for oxidative detoxification [38,41]. Trehalose 6-phosphate synthetase (TPS) is a crucial enzyme in trehalose synthesis. The expression of TPS3 can be induced by various stress conditions, such as ABA, IAA, salt, drought, and cold [42]. In the present study, six genes (Seita.1G347300, Seita.2G197800, Seita.5G305600, Seita.5G318400, Seita.6G144600, and Seita.6G166700) were found to be involved in trehalose biosynthesis. This suggests that these specific genes play a role in the response of foxtail millet to freezing stress. In this study, 31 DEGs were significantly enriched in the “starch and sucrose metabolism” pathways. The soluble sugar content of foxtail millet leaves increased under freezing treatment (Figure 3A). Previous studies have confirmed that cold resistance is positively correlated with starch and sucrose contents [38,43].

The regulation of starch metabolism in plants involves a series of enzymes, with β-amylase (EC 3.2.1.2; BAM) playing a crucial role in the accumulation of soluble sugars under low-temperature stress. The findings of this study indicate that the expression of BAM (Seita.9G317500 and Seita.9G544600) was upregulated during freezing stress, leading to an increase in β-amylase activity and sucrose content. Therefore, we propose that β-amylase activity promotes starch degradation in response to freezing stress (Appendix A). This is consistent with a previous study that demonstrated the positive impact of PtrBAM1 overexpression in tobacco [44], which led to increased BAM activity, promoted starch degradation, and ultimately increased soluble sugar content. Similarly, silencing AtBAM3 further validates its function in this process [45]. According to research, cold stress can result in an excess amount of starch in chloroplasts [46]. Typically, starch is synthesized in chloroplasts, then converted to triose phosphate, transported to the cytoplasm, and used to make soluble sugars like sucrose and maltose [47]. However, when exposed to cold stress, the transport of trisaccharide phosphate to chloroplasts is inhibited owing to photorespiration inhibition, leading to the accumulation of starch in chloroplasts [48]. This phenomenon has been observed in millet plants.

### 3.3. BR Signal Network Responses to Freezing Stress

BRs regulate membrane structure and growth by binding to membrane protein receptors [17,49]. Exogenous BR treatment has been found to increase the antioxidant defense and osmoregulatory capacity of plants under low-temperature stress [50], indicating that BRs play an important role in alleviating stress in crops [51,52]. In this study, we examined DEGs related to BR biosynthesis, catabolism, and signaling pathways. The expression of BR resistance (BZR1/2; Seita.5G143700) in response to freezing stress was comparable to that observed in Arabidopsis. Overexpression of BZR1 and knockdown of BIN2 increased freezing resistance via both CBF-dependent and CBF-independent pathways [53]. BRI1, a transmembrane receptor kinase, plays a crucial role in sensing and transducing signals related to membrane-surface BRs. This study found that freezing stress can cause an increase in the expression of the brassinosteroid insensitive 1 (BRI1) precursor (Seita.1G085300, Seita.1G085800, Seita.2G377800, Seita.3G240900, Seita.4G251500, Seita.5G432500, Seita.6G205500, Seita.9G227100, Seita.9G429400, Seita.J023100, and Setaria_italica_newGene_3251,) in the hormone signal transduction pathway. This increase in BRI1 expression activates the brassinolide signal transduction pathway. Recent studies have shown that BAM acts as a transcription factor and interacts with the BR signal transduction pathway, which may be a conserved mechanism among various plants [54]. In this study, we identified eight AMY genes and two BAM genes that were upregulated in millet leaf tissue. We hypothesized that BRs regulate plant osmotic pressure and improve plant freezing tolerance by regulating AMY and BAM activities under freezing stress, inducing starch degradation, and increasing sugar content in leaves.

### 3.4. TFs Responding to Freezing Stress

Under low-temperature stress, plants activate transcription factors (TFs) via signal transduction pathways to enhance cold resistance by inducing the downstream expression of resistance-related genes [55,56].

AP2/ERF genes are part of a plant-specific TF family that plays a role in stress response regulation [57]. In *Arabidopsis*, the AP2/ERF transcription factor HcTOE3 has been found to positively regulate frost tolerance. The transgenic strains showed improved survival rates; increased antioxidant enzyme activity and proline content; and decreased malondialdehyde content, electrolyte extravasation, and accumulation of ROS. This study identified 27 differentially expressed AP2/ERF genes that could serve as potential candidates for further investigation into the freezing reaction mechanism of millet. These genes have been shown to play a role in the response to different types of abiotic stress via phytohormones such as ABA, ET, GA, CTK, and BR [58,59].

The regulation of plant development and stress response is crucial and is influenced by gene families such as MYB, WRKY, bHLH, and NAC [60,61,62,63]. Research has shown that the overexpression of MYB96 in *Arabidopsis* can enhance plant freezing tolerance by regulating the target gene, lipid transporter LTP3 [64]. Similarly, the overexpression of LcMYB4 in L. chinensis can increase soluble sugar content and the expression of cold-induced genes, which in turn improves the cold tolerance of transgenic plants [65]. The VbWRKY32 transcription factor found in verbena has been shown to have the positive regulatory effect of increasing the transcription levels of cold-responsive genes [66]. Additionally, the inducer of C-repeat/dehydration responsive element binding factor (CBF) expression (ICE), a bHLH transcription factor, plays a role in the low-temperature signaling pathway. Another study found that the tobacco transcription factor gene NtbHLH123 is involved in plant temperature regulation, and the overexpression of this gene enhances the activity of antioxidant enzymes, ultimately improving cold tolerance in tobacco [67]. One study found that the expression of the SsNAC23 gene was significantly induced at 4 °C in sugarcane [68].

Transcriptome results also revealed that other glutamic transcription factors, such as AP2/ERF, MYB, WRKY, NAC, C2H2, and the bHLH family, were induced with freezing treatment, indicating their significant role in the response to freezing stress. Previous studies have shown that these members of the transcription factor family can significantly affect freezing tolerance in plants; however, further research is needed to determine whether these TFs act independently or synergistically to enhance freezing tolerance in foxtail millet.

## 4. Materials and Methods

### 4.1. Plant Materials and Freezing Treatment

Huangjinmiao (H) was used in all experiments. H seeds were provided by Chifeng Academy of Agricultural and Animal Husbandry Sciences, China, and were first air-dried before being planted in the experimental greenhouse of the Crop Chemistry Control Centre of Shanxi Agricultural University, located in Taigu County, Shanxi Province, China. The greenhouse maintained a temperature of 25–28 °C and a day/night cycle of 12 h/12 h. When the second leaf was fully unfolded, it was placed in a freezer at −4 °C for the freezing stress experiment. After treatment for seven different durations (0, 2, 4, 6, 8, 10, and 12 h), the excised leaves were collected and immediately frozen in liquid nitrogen. Three biological replicates were used for each time point. In total, 21 leaf samples were collected and stored at −80 °C for further analysis.

### 4.2. Determinations of Leaf Physiological Indicators

#### 4.2.1. Growth Index Measurement

A ruler was used to measure absolute plant height from the neck of the ground root to the tip of the plant in centimeters (cm). After treatment, growth was recovered for 7 days, and the survival rate was determined as: number of living plants/original total × 100%.

#### 4.2.2. Physiological Indicators

To determine SOD activity, we used the nitroblue tetrazolium chloride (NBT) photochemical reduction method. A 0.1 g sample of the measured leaves was placed in a mortar with 2 mL of 50 mM phosphate buffer (pH 7.8) and added to an ice bath. The slurry was ground, put into a centrifuge tube, and centrifuged at 4 °C and 10,000 r·min^−1^ for 20 min. The extracted supernatant was an SOD, POD, and CAT enzyme solution. A 100 μL sample of the SOD enzyme solution was divided into two groups: one with 5 mL of NBT reaction solution (experimental group) and one with the same amount of distilled water (blank group). The enzyme solution treatment was placed at 25 °C for approximately 20 min, and the blank treatment was placed in the dark. As a reference, the color was compared at 560 nm, and the absorbance value was recorded.

To determine POD activity, 50 μL of enzyme solution was added to 3 mL of reaction solution (using the guaiacol method), and the absorbance was measured at 470 nm for 3 min.

To determine CAT activity, 100 μL of the enzyme solution was taken in a quartz colorimetric cup, to which we added 1.5 μL of pH 8.7 phosphate buffer, 1 mL of distilled water, and 0.3 mL 0.1 M H_2_O_2_. Absorbance was measured at 240 nm and recorded every 30 s for 3 min.

The thiobarbituric acid (TBA) colorimetric method was used to determine MDA content. Leaves weighing 0.4 g were crushed. The volume was adjusted to 5 mL with 0.1% trichloroacetic acid (TCA). This was followed by 5 mL 0.5% thiobarbituric acid. After boiling for 15 min in a water bath, the mixture was immediately placed in an ice bath and cooled to room temperature. Then, it was transferred to a 10 mL centrifuge tube and centrifuged at 3000 r·min^−1^ for 15 min. The supernatant was collected, and its volume was measured. The absorbance values at 532 and 600 nm were measured using 0.5% thiobarbituric acid as reference.

The REC of the seedling exudate was determined using a DDS-11A conductivity meter [69]. The second leaf of the foxtail millet seedlings was washed three times with distilled water. After wiping dry with gauze, 0.1 g leaves were weighed and cut into 0.5 cm segments. The segments were placed in a calibration tube containing 10 mL of deionized water and covered. After soaking for 12 h at room temperature, the conductivity of R1 was measured using a conductivity meter. After boiling for 15 min in a water bath, the mixture was immediately placed in an ice bath and cooled to room temperature. The conductivity of the extract, R2, was measured.

Free proline content was determined using the acid–ninhydrin method [70]. Leaves (0.2 g) were ground, and the volume was adjusted to 5 mL with 3% sulfosalicylic acid. The solution was boiled for 10 min, transferred to a 10 mL centrifuge tube after cooling, and centrifuged at 4000 r·min^−1^ for 10 min; the supernatant was the solution to be tested. A 2 mL sample of the supernatant was placed in a stoppered test tube, to which 2 mL of glacial acetic acid and 2 mL of acidic ninhydrin were added. After shaking, the mixture was placed in a boiling water bath for 30 min. After cooling to room temperature, toluene (5 mL) was added, and the mixture was extracted in the dark. The absorbance at 520 nm was measured using 1 mL of distilled water and 1 mL of glacial acetic acid + 1.5 mL acidic ninhydrin as references.

Soluble protein determination was performed using Coomassie Brilliant Blue G-250 staining [23]. Leaves (0.2 g) were ground, and the resulting mixture was diluted to a volume of 10 mL using distilled water. The 2–3 mL solution was centrifuged in a centrifuge tube at 4000× *g* for 10 min. The supernatant was extracted, and 0.5 mL distilled water and 5 mL Coomassie Brilliant Blue G-250 reagent were added. After mixing, the mixture was incubated for 2 min, and the absorbance was measured at 595 nm.

For the determination of soluble sugar content, 0.5 g foxtail millet leaves were accurately weighed using the 3,5-dinitro salicylic acid method [23]. Leaves were combined with 5 mL of 80% ethanol, ground, mixed well, and placed in an 80 °C water bath for 30 min. During this period, the reaction vessel was stirred, cooled, and centrifuged at 3500× *g* for 10 min. The supernatant was transferred to a 25 mL volumetric flask and diluted with 80% ethanol, which was the extract. The extract, distilled water, and anthrone-H_2_SO_4_ (2 mL each) were placed in test tubes and heated in a water bath for 2 min. After cooling, absorbance was measured at 630 nm. The residue after the extraction of soluble sugars was washed with methanol, and starch content was determined by enzymatic hydrolysis. Physiological indicators such as starch, sucrose, and β-amylase activity [71] were compared between the control and stress treatment groups. The determination of these indicators was conducted using an ELISA Kit (Michy, Suzhou, China) with three replicates and a microplate reader.

After the samples were collected, the levels of BR content were measured with an enzyme-linked immunoassay (ELISA) [72].

#### 4.2.3. Transmission Electron Microscopy

The seedlings treated at −4 °C for 8 h were selected, and the leaves of the treatment group and control group (fast, accurate, small) were cut and placed in 2.5% glutaraldehyde solution at 4 °C for 24 h, rinsed with 0.1 mol/L phosphoric acid rinse solution at room temperature for 15 min (three times), and then placed in a 1% osmic acid refrigerator at 4 °C for 2 h. Samples were then washed with ddH_2_O for 15 min three times and dehydrated with 100% propylene oxide for 5 min (three times). Next, they were infiltrated in propylene oxide + embedding solution (2:1) for 2 h, in propylene oxide + embedding solution (1:2) for 3 h or overnight, in pure embedding solution overnight, and in pure embedding solution at room temperature for 3–4 h. Then, the sample block was placed in the embedding template; after adding the pure embedding agent, polymerization was carried out in a constant-temperature oven under the following conditions: 37 °C for 12 h, 45 °C for 12 h, and 60 °C for 48 h. Finally, samples were cut using semi-thin (1 μm) positioning with a Leica EMUC 6 (German Lycra) ultra-thin slicer (70 n) and double-stained with 3% uranium acetate-lead citrate for analysis. The samples were then imaged using a JEM1230-type transmission electron microscope (Japanese JEOL company, Tokyo, Japan).

### 4.3. RNA Extraction, Library Preparation, RNA-Seq, and Sequence Assembly

A TRIzol kit (Accurate Biology, Changsha, China) was used to extract the total RNA from each leaf sample. Total RNA was assessed using a NanoDrop 2000 spectrophotometer (ThermoFisher Scientific, Wilmington, DE, USA) and an Agilent Bioanalyzer 2100 system (Agilent Technologies, Palo Alto, CA, USA) to assess total RNA quality, integrity, and quantity. Sequencing libraries were constructed from high-quality RNA samples. After purification, evaluation, and clustering, libraries were sequenced using the Illumina HiSeq 2500 platform (Illumina Inc., San Diego, CA, USA). The high-quality sequences were aligned to the *Setaria italica* reference genome using HISAT2 v2.2 (http://ccb.jhu.edu/software/hisat2/index.shtml, accessed on 1 December 2021) [24]. The mapped reads were assembled using String Tie v2.1.5 (https://ccb.jhu.edu/software/stringtie/index.shtml, accessed on 1 December 2021) [25]. Differential gene expression was analyzed using DESeq2 (http://www.bioconductor.org/packages/release/bioc/html/DESeq2.html, accessed on 1 December 2021) [25] and calculated as FPKM [73] values. Genes with fold change (FC) of ≥1.5 and false discovery rate (FDR) of <0.05 were considered differentially expressed. Gene function annotation was performed using BLAST (http://blast.ncbi.nlm.nih.gov/Blast.cgi, accessed on 1 December 2021) [73] in two databases: Gene Ontology (http://www.geneontology.org/, accessed on 1 December 2021) [74] and the Kyoto Encyclopedia of Genes and Genomes (http://www.genome.jp/kegg/, accessed on 1 December 2021) [75].

### 4.4. qRT–PCR

To test the reliability of transcriptome sequencing with qRT–PCR, we selected 15 DEGs. RNA was extracted from leaves using a TRIzol kit (Accurate Biology, Changsha, China); cDNA was prepared using a reverse transcription kit (Accurate Biology, Changsha, China); and real-time PCR was performed using the SYBR Green method (Accurate Biology, Changsha, China). SiActin (SETIT_026509mg) was used as an internal standard, and relative expression levels were calculated using the 2^−ΔΔCt^ method [76]. At least three replicates were performed in each independent experiment. Primer Premier 5 software (Primer Premier 5.0) was used to design the PCR primers (Appendix A).

### 4.5. Statistical Analysis

Statistical data were analyzed using Microsoft Office Excel 2010 and a Data Processing System, SPSS version 22.0 (IBM Corp., Armonk, NY, USA). Analysis of variance (ANOVA) followed by Tukey’s multiple comparison test. (*p* < 0.05).

## 5. Conclusions

The response of foxtail millet to freezing stress was analyzed at the morphological, physiological, and transcriptomic levels. The REC and MDA levels increased significantly, suggesting that freezing stress may be associated with cell-membrane damage. Freezing stress experienced by foxtail millet leads to the upregulation of genes related to sucrose and starch metabolism, as well as BR signaling transduction. Additionally, there was a significant increase in soluble sugar content. These changes in gene and material content suggest that foxtail millet utilizes osmotic regulation in response to freezing stress (Figure 11). The findings of this study shed light on the response mechanism of foxtail millet to freezing stress and offer valuable insights for future research on the environmental adaptability of this crop.

## Figures and Tables

**Figure 1 ijms-24-11590-f001:**
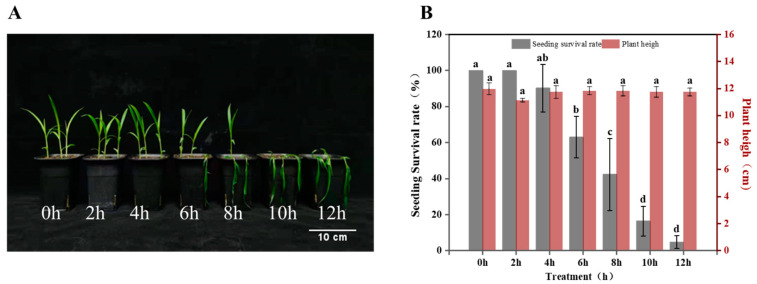
Morphological changes (**A**) and leaf apparent morphology indices (**B**) of foxtail millet (variety H) under −4 °C freezing stress. Data are means ± standard deviations (SDs) (*n* = 3). Different letters indicate significant differences at *p* < 0.05.

**Figure 2 ijms-24-11590-f002:**
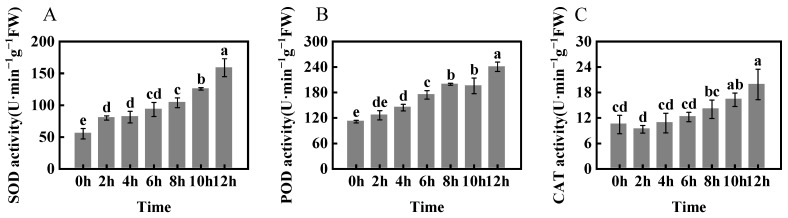
Changes in SOD (**A**), POD (**B**), and CAT (**C**) activities in foxtail millet (variety H) under −4 °C freezing stress. Data are means ± SDs (*n* = 3). Different letters indicate significant differences at *p* < 0.05.

**Figure 3 ijms-24-11590-f003:**
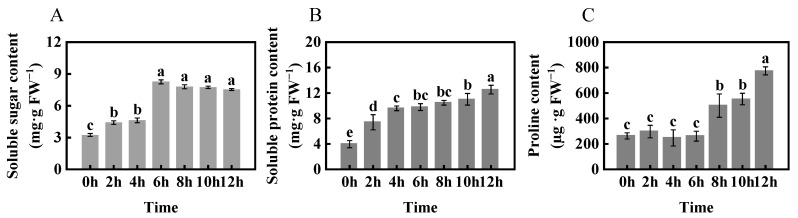
Changes in soluble sugar content (**A**), soluble protein content (**B**), and free proline content (**C**) of foxtail millet (variety H) under −4 °C freezing stress. Data are means ± SDs (*n* = 3). Different letters indicate significant differences at *p* < 0.05.

**Figure 4 ijms-24-11590-f004:**
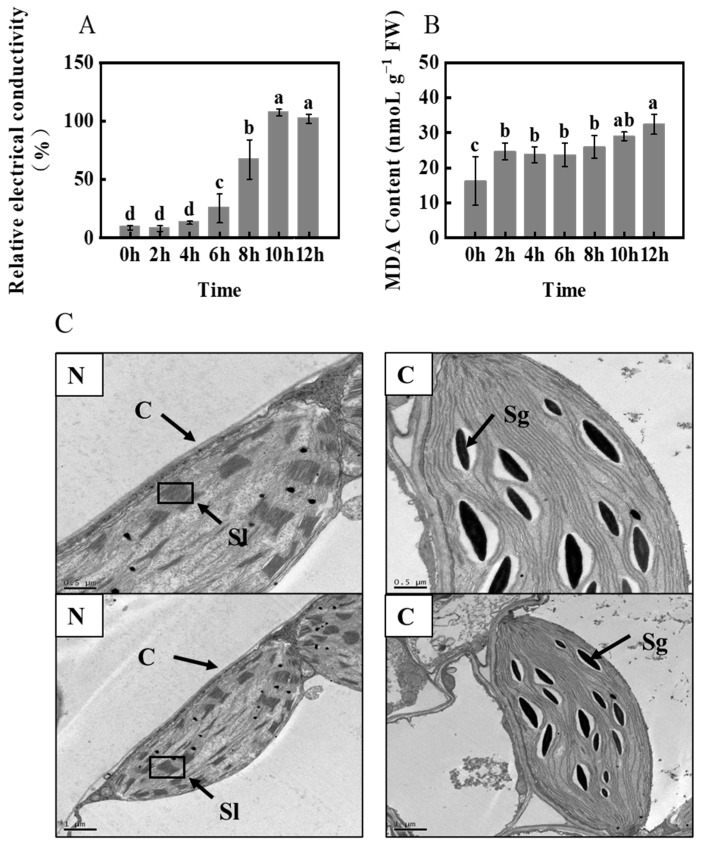
REC (**A**), MDA content (**B**), and chloroplast ultrastructures (**C**) of foxtail millet (variety H) under −4 °C freezing stress. Sl: stroma lamella; C: chloroplast; Sg: starch granules. Data are means ± SDs (*n* = 3). Different letters indicate significant differences at *p* < 0.05.

**Figure 5 ijms-24-11590-f005:**
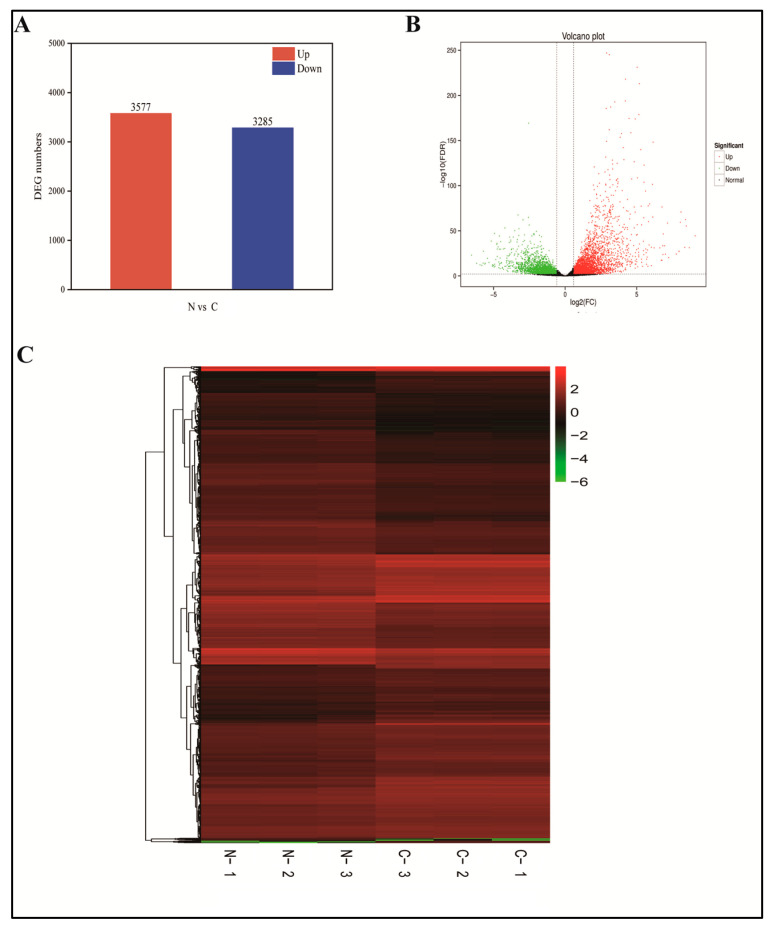
Transcriptome analysis of DEGs in the N and C treatments of foxtail millet. DEGs in the N and C treatments (**A**). Volcano plot showing DEGs in the N and C treatments of foxtail millet (**B**). Heat maps representing DEGs expression profiles after freezing treatment (**C**). To determine the significance of DEGs, a threshold of q < 0.05 was used. The grey points represent transcripts that were not significantly changed in the N library in comparison with the C library. N: normal; C: frozen.

**Figure 6 ijms-24-11590-f006:**
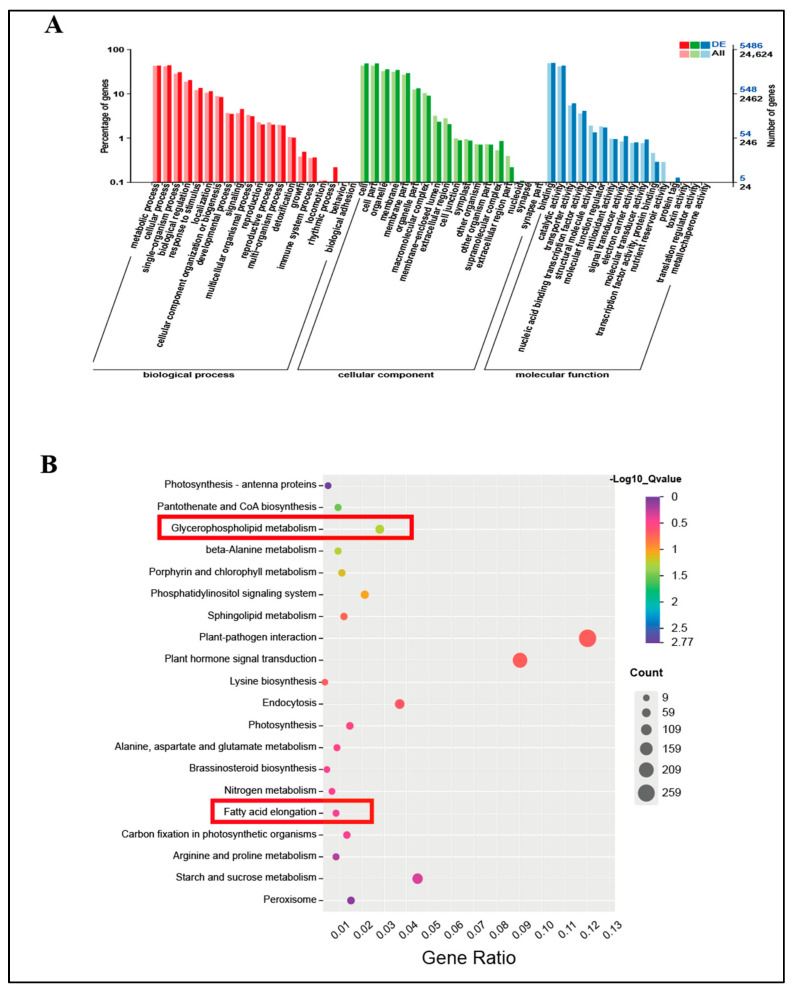
GO enrichment analysis of the 30 most abundant DEGs in foxtail millet leaf after N and C treatments (**A**). KEGG enrichment analysis of the 20 most abundant DEGs in foxtail millet (**B**). GeneRatio = number of differentially expressed genes enriched in this pathway/number of all differentially expressed genes enriched in the KEGG database. The red boxed portion of the KEGG pathway is associated with membrane damage.

**Figure 7 ijms-24-11590-f007:**
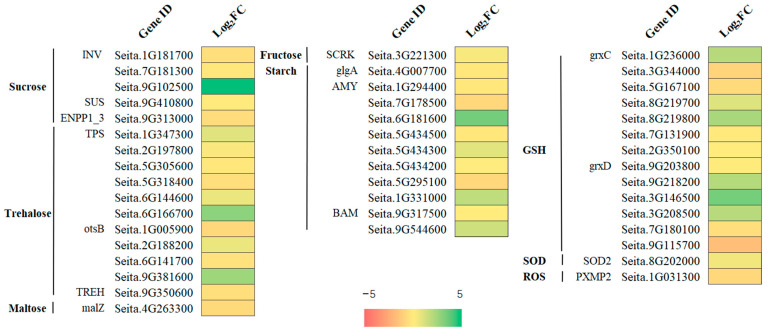
Heat map of the main DEGs involved in starch and sucrose metabolism, and the antioxidant defense system in foxtail millet seedlings.

**Figure 8 ijms-24-11590-f008:**
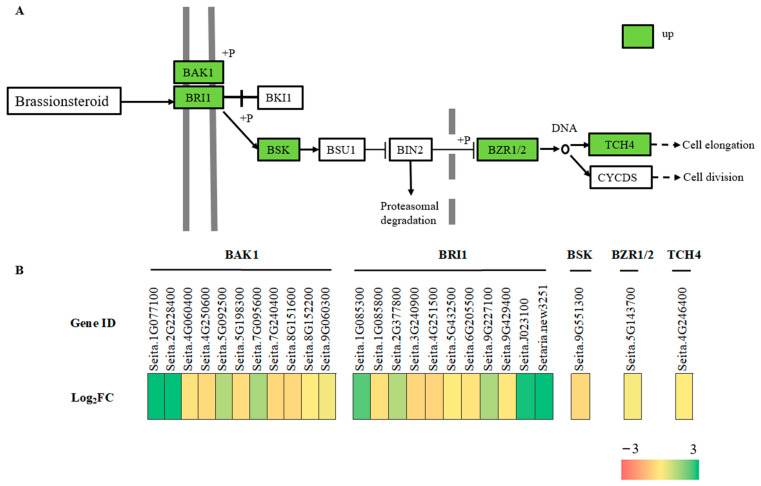
Freezing-induced changes in gene expression levels in the BR signaling network revealed by N vs. C comparisons. (**A**) Gene expression pattern in the BR signaling pathway. (**B**) Heat map of DEGs involved with BRs.

**Figure 9 ijms-24-11590-f009:**
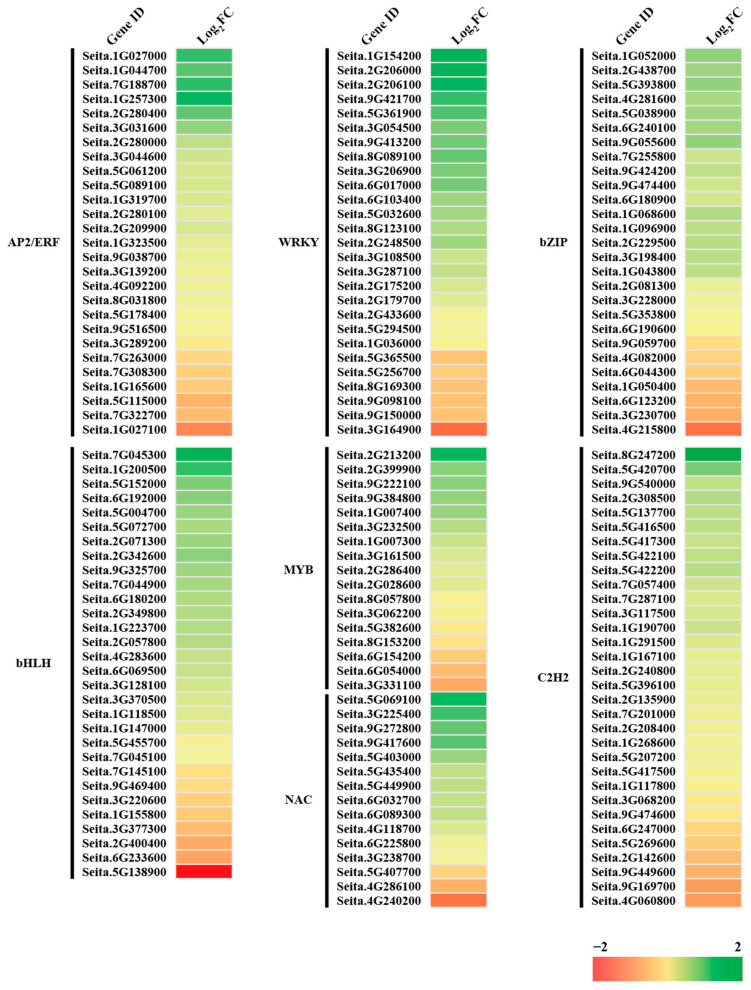
Heat map of the main transcription factors.

**Figure 10 ijms-24-11590-f010:**
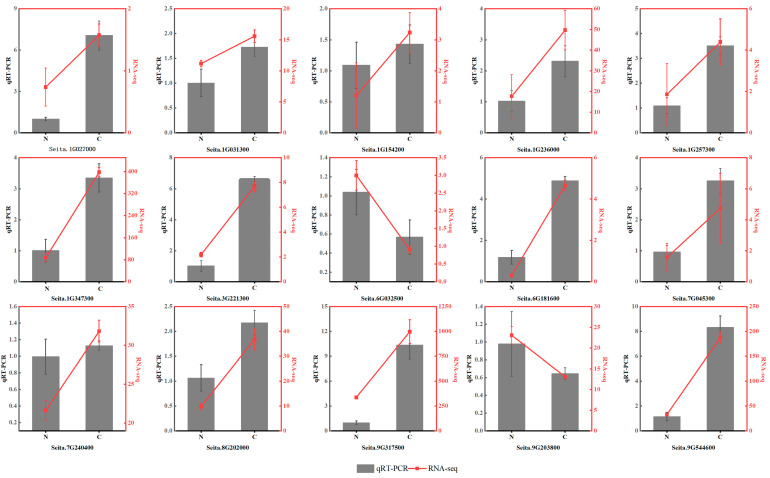
Selected DEG genes detected with qRT–PCR in the leaves of foxtail millet under freezing treatment. Data are means ± SDs (*n* = 3). N: normal; C: frozen.

**Figure 11 ijms-24-11590-f011:**
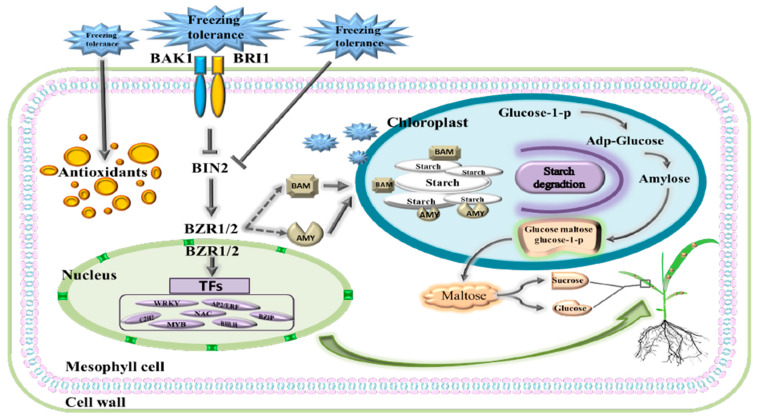
Proposed model of foxtail millet response to freezing stress. Under freezing stress, the cell membrane of foxtail millet senses external frozen signals, which are then transmitted to the nucleus by antioxidants. The BR signal, starch and sucrose metabolism, and many related transcription factors are induced to positively regulate plant freezing tolerance. Freezing damage results in the accumulation of starch, which in turn releases a portion of maltose. This maltose is then metabolized into sucrose and free hexose. Sucrose is subsequently transported to the leaves, where it enhances freezing resistance. Solid and broken arrows indicate activation and speculative regulation, respectively, whereas lines ending with a bar show negative regulation.

**Table 1 ijms-24-11590-t001:** Alignment statistics calculated with the reference gene.

Sample	Clean Reads	Clean Bases	GC Content	% ≥ Q30	Mapped Reads	Uniq Mapped Reads
N-1	36,475,245	10,900,981,318	53.90%	94.82%	94.76%	91.27%
N-2	38,642,791	11,547,424,178	53.67%	94.75%	94.65%	91.17%
N-3	38,450,107	11,483,404,412	53.33%	94.06%	94.46%	91.32%
C-1	28,449,360	8,503,541,058	52.94%	94.65%	95.00%	92.38%
C-2	28,701,001	8,585,680,730	53.39%	95.03%	95.22%	92.54%
C-3	37,159,829	11,117,086,398	52.79%	94.84%	94.65%	92.12%

N: normal; C: frozen.

## Data Availability

Data supporting the discovery of our work are available within the paper and its Appendix A.

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
