# Peer review of "Transcriptome Analysis Reveals Brassinolide Signaling Pathway Control of Foxtail Millet Seedling Starch and Sucrose Metabolism under Freezing Stress, with Implications for Growth and Development"

_ijms, 2023, doi:10.3390/ijms241411590_

Round 1

Reviewer 1 Report

Authors investigate the physiology and transcriptome of foxtail millet exposed to below-zero temperature. The author performed analyses covering several parameters like the seedling's survival rate, the activity of antioxidant enzymes ( SOD, POD, and CAT), changes in soluble sugar content, soluble protein content, and free proline content as well as relative electrical conductivity, MDA content, and chloroplast ultrastructure. Authors performed also transcriptomic analysis and identify genes differentially expressed during stress treatment. The manuscript contains the results of many experiments, however, all obtained results are conformation of well -investigated effects of freezing stress commonly found in plants, thus they are lacking scientific novelty. What is more, the manuscript is written very chaotically and inconsistently.

 Some section headings do not correspond with their content. For example: In section 2.1 which is entitled “ Leaf Apparent Morphology Indices of Foxtail Millet During Freezing Treatment” authors present mainly seedling survival rates and no leaves morphological observations are present.  Section  3.1 is entitled “Effects of Freezing Stress on Membrane Lipid Metabolism of Foxtail Millet Leave”, but the authors did not investigate lipid metabolism but a degree of membrane damage.

Section 2.9. Phytohormone Related DEGs are described very generally and unclearly. Regardless of the unclear description, the authors over-interpret the obtained results. (“KEGG analysis of the DEGs in foxtail millet leaves under freezing stress showed that freezing stress significantly affected hormone signal transduction”( p.9 lines 228-229). Observed by authors changes in gene expression are not evidence of affected hormone signal transduction.

The discussion is not only very general but in many points conclusions are unsupported by the results.  For example “These DEGs were enriched in ‘‘fatty acid elongation (ko00564)’’ and “glycerophospholipid metabolism (ko00062)’’. In response to freezing stress, genes in the cells of foxtail millet seedlings were influenced by increased levels of unsaturated fatty acids and membrane fluidity. This response may serve as a mechanism of cellular self-repair (p. 12; 280-284)”. The authors investigate only relative electrical conductivity and MDA content. MDA level is a commonly known marker of oxidative stress and REC indicates the relative percentage of cell membrane leakage. These are both indicators of membrane damage. There is no information about levels of unsaturated fatty acids and membrane fluidity in the manuscript. What links fatty acid elongation and glycerophospholipid metabolism and membrane fluidity? Which genes from ko00564 and ko00062 were differently regulated? Were they upregulated or downregulated?

In 3.2. section  “Metabolic Pathways of Starch and Sucrose Under Freezing Stress” the authors claim that h β-  amylase plays a crucial role in the accumulation of soluble sugars under low-temperature stress since its activity promotes starch degradation and that genes encoding β-  amylases were  upregulated during freezing stress. However, authors observed an accumulation of starch grains in chloroplast exposed to low temperature. The explanation for this contradictory observation is not clear.  Authors should also remember that not genes but enzymes encoded by these genes are involved in an enzymatic reaction, so the sentence “genes related to BAM promote starch degradation” (p.13; line 314) and all similarities should be rewritten.

And many others…

In conclusion. Due to the low scientific novelty of obtained results and many conclusions are not supported by the results the manuscript should not be published. 

The language is difficult to understand. Inappropriate vocabulary is often used.

Reviewer 2 Report

The manuscript by Zhao and co-authors tries to integrative the molecular mechanisms response to freezing stress in foxtail millet. They analyzed the morphological, physiological, and transcript levels in the seedlings and examined free proline, soluble sugar, and soluble protein contents. Illumina-based RNA-Seq identified 6,862 DEGs, and subsequent bioinformatics approaches characterized them by using GO- and KEGG pathway-enrichment analysis. The results suggested that the stress led changes in transcript levels related to starch/sucrose metabolism and BR signaling transduction. The expression patterns of several key genes were verified by qRT-PCR methods. The study documents some interesting facts concerning the freezing stress responses in the species. The experimental strategies are well designed. However, there are several places of the manuscript that need to be revised as follows.

Methods:

1)     The authors should add more explanation of the grain reference genome. What are the name and assembled version?

2)     Please describe all the software/databases used in this study. For example, GO- and KEGG pathway enrichment analysis. Also, would you please add all the parameters in your data analyses?

3)     Please cite the latest paper about all the software/databases. For example, KEGG.

4)     You mentioned: L479, “Genes with FC (fold change) of ≥ 1.5 and…” Why did you focus on significant up-regulated genes? How about down-regulated genes?

5)     Regarding the heatmaps (e.g., Fig. 5C), I think that use of FPKM values is not ideal where cross-sample differential expression analysis. See, Dillies et al. 2013 https://pubmed.ncbi.nlm.nih.gov/22988256/.

Data availability:

The authors should deposit your read data by an Illumina platform in public repositories (e.g., NCBI SRA, https://www.ncbi.nlm.nih.gov/sra).

Figures and Tables:

1)     Fig. 1A: Do you have any scale bars in the experiment?

2)     Fig. 1B, Fig. 2, and Fig. 3: What are the letters (e.g., a, b, c, and d)? The authors should describe the name of statistical tests with p-values.

3)     Fig. 5C, Fig. 7, Fig. 8, and Fig. 9: What are the relative expressions? FPKM or Z-score?

4)     Fig. 6: What is “Gene Ratio”?

5)     Fig. 10: Do the bar plots represent averaged abundance or median? The authors should describe the error bars, experimental replicates, and statistical tests.

Minor comments/suggestions/errors
The manuscript needs extensive language editing. There were many typos.

See above.

Round 2

Reviewer 1 Report

The manuscript was significantly improved.  All necessary corrections have been made. In my opinion, the manuscript can be published in the present form. 

Author Response

Thank you for dedicating your time to review my revised draft. I greatly appreciate your valuable comments and professional suggestions, which have significantly enhanced the quality of the article.